# Oral Microbiome: A Review of Its Impact on Oral and Systemic Health

**DOI:** 10.3390/microorganisms12091797

**Published:** 2024-08-29

**Authors:** John J. Rajasekaran, Hari Krishnan Krishnamurthy, Jophi Bosco, Vasanth Jayaraman, Karthik Krishna, Tianhao Wang, Kang Bei

**Affiliations:** 1Vibrant Sciences LLC, Santa Clara, CA 95054, USA; hari@vibrantsci.com (H.K.K.); vasanth.jayaraman@vibrantsci.com (V.J.); karthik@vibrantsci.com (K.K.); tianhao.wang@vibrantsci.com (T.W.); kang@vibrantsci.com (K.B.); 2Vibrant America LLC, Santa Clara, CA 95054, USA; jophi.b@vitasoft-tech.com

**Keywords:** oral microbiome, oral diseases, systemic diseases, dysbiosis, probiotics, oral microbiota, oral care

## Abstract

Purpose of review: This review investigates the oral microbiome’s composition, functions, influencing factors, connections to oral and systemic diseases, and personalized oral care strategies. Recent findings: The oral microbiome is a complex ecosystem consisting of bacteria, fungi, archaea, and viruses that contribute to oral health. Various factors, such as diet, smoking, alcohol consumption, lifestyle choices, and medical conditions, can affect the balance of the oral microbiome and lead to dysbiosis, which can result in oral health issues like dental caries, gingivitis, periodontitis, oral candidiasis, and halitosis. Importantly, our review explores novel associations between the oral microbiome and systemic diseases including gastrointestinal, cardiovascular, endocrinal, and neurological conditions, autoimmune diseases, and cancer. We comprehensively review the efficacy of interventions like dental probiotics, xylitol, oral rinses, fluoride, essential oils, oil pulling, and peptides in promoting oral health by modulating the oral microbiome. Summary: This review emphasizes the critical functions of the oral microbiota in dental and overall health, providing insights into the effects of microbial imbalances on various diseases. It underlines the significant connection between the oral microbiota and general health. Furthermore, it explores the advantages of probiotics and other dental care ingredients in promoting oral health and addressing common oral issues, offering a comprehensive strategy for personalized oral care.

## 1. Introduction

The human oral cavity harbors a diverse and complex microbial community, collectively known as the oral microbiome [1]. It has diverse microorganisms, including bacteria, archaea, fungi, and viruses [2]. It exists in the form of a biofilm and contributes significantly to maintaining oral homeostasis by fostering a harmonious balance within the oral environment [1]. Through its natural defense mechanisms, the oral microbiome acts as a protective barrier, preventing the colonization of harmful pathogens and preventing potential infections in the oral cavity [3]. Furthermore, a balanced oral microbiome is instrumental in preventing the development of oral diseases, such as dental caries and periodontal diseases, as it helps control the growth and activity of pathogenic bacteria [1,3]. Understanding the diverse roles of the oral microbiome is essential for devising effective preventive and therapeutic strategies to promote oral health.

Interestingly, various civilizations have shown an extensive understanding of the vitality of oral health in the past. Remarkably, in ancient Egypt, the arid climate and special burial customs led to the preservation of well-kept skeletons and mummies. Researchers have learned a great deal about the historical dental practices of ancient civilizations by examining these remnants [4]. In antiquity, societies likely employed wooden toothpicks, chew sticks, tree twigs, and animal bones as implements for oral hygiene. Impressively, traces of caries and periodontal disease date back thousands of years to the upper Paleolithic era. As opposed to their prevalent occurrence in current times, dental caries were notably infrequent in ancient Egypt due to a lack of fermentable carbohydrates in their diet, impeding plaque buildup. Additional factors, such as tooth wear, including occlusal and interproximal wear, further contributed to the scarcity of caries by smoothing surfaces and discouraging plaque proliferation [4]. The increased availability and consumption of sugar-rich foods and beverages over the past few decades has resulted in a substantial shift in dietary patterns and an increase in dental problems. The rise in metabolic illnesses such as obesity, non-alcoholic fatty liver disease, cardiovascular disease, and diabetes is partly attributable to this increased consumption of sugars [5].

This historical perspective underscores the importance of dietary choices and their impact on oral and overall health.

Under healthy conditions, the oral microbiome thrives in a favorable commensal association with its environment, much like other body regions, including the skin, gut, or vagina [6]. However, it is important to acknowledge that under certain circumstances, certain opportunistic microorganisms within the oral microbiome can undergo a shift, becoming harmful pathogens. This transformation can have implications for the development of various oral and systemic diseases [6,7]. Major oral diseases such as dental caries a/nd periodontal diseases are both caused by and contribute to disruptions in the oral microbiota balance, known as dysbiosis. This dysbiosis can extend its effects to various chronic systemic diseases, leading to the initiation or worsening of conditions such as metabolic diseases, CVD, respiratory diseases, rheumatoid arthritis (RA), adverse pregnancy outcomes, IBD, AD, autism spectrum disorders, and oral mucosal diseases [8].

In the past decade, significant progress has been made in understanding the impact of the oral microbiome on health and disease, propelled by innovative genomic technologies such as Next-Generation Sequencing (NGS) and advanced bioinformatic tools. NGS is an umbrella term that encompasses various methods like 16S rRNA sequencing, metagenomics, shotgun metagenomics, and quantitative Real-Time PCR (RT-PCR), all of which have replaced conventional culture-based techniques in unravelling the complexities of the oral microbiome [9,10]. In this context, stimulated saliva collection offers a controlled and comprehensive method for investigating the oral microbiome, providing valuable insights into its composition, functions, and implications for human health. (Detailed techniques for the analysis of the oral microbiome are provided in Appendix A ([1,11,12,13,14,15,16,17,18,19,20,21,22]).) It is important to acknowledge that the oral microbiome is molded by various factors, such as diet, oral hygiene practices, smoking, systemic illnesses, and changes in the oral environment [23]. Therefore, emphasizing the importance of promoting good oral hygiene practices, adopting a healthy diet, and making positive lifestyle changes can play a pivotal role in nurturing a balanced oral microbiome ecosystem.

This review extensively examines the diverse microbial composition and its role in maintaining oral equilibrium, outlining the significance of various bacterial genera. Additionally, it delves into the oral–gut axis and microbial translocation, unravelling the interconnection between the mouth and gut. We also explore how dysbiosis can instigate a cascade of consequences, not only leading to various oral diseases such as dental caries, periodontitis, halitosis, and oral candidiasis but also exerting a profound impact on systemic health, encompassing conditions like gastrointestinal diseases, CVD, diabetes, neurological diseases, and autoimmune diseases. Lastly, understanding the pivotal role of daily dental care practices is essential for maintaining oral hygiene and preventing oral diseases, thus contributing to the preservation of a balanced oral microbiome ecosystem and good systemic health.

Search methods: To provide a comprehensive review of the oral microbiome, we conducted an extensive literature search. We used the following keywords: “Oral Microbiome”, “Oral Diseases”, “Systemic oral diseases”, “oral care”, “dental practices”, “oral probiotics”, and many more. Our search spanned multiple databases including PubMed, Google Scholar, and Web of Science.

Inclusion and exclusion criteria: This review paper includes case–control studies, cross-sectional studies, retrospective and prospective cohort studies, and randomized controlled trials that examined the composition of, role of, and factors that influence oral microorganisms, connections between oral and systemic diseases, and personalized oral care strategies.


*Inclusion criteria:*
Peer-reviewed articles;Research focusing on the oral microbiome and related disease conditions;Articles in English.

*Exclusion criteria:*
Non-peer-reviewed articles;Studies focusing on non-oral microbiomes;Articles not available in English.


This thorough methodology ensures that our review encompasses a broad and relevant range of studies, providing a robust overview of the current state of knowledge in this field.

## 2. The Composition and Role of the Oral Microbiome

With over 700 known bacterial species, the oral microbiome is the second most diverse microbiota within the human body [1]. Principally, this vibrant ecosystem encompasses a wide range of bacterial phyla, including Actinobacteria, Bacteroidetes, Chlamydia, Euryarchaeota, Fusobacteria, Firmicutes, Proteobacteria, Spirochaetes, and Tenericutes [24]. In a healthy oral cavity, the prevailing bacterial genera primarily encompass both Gram-positive and Gram-negative species. Within the realm of Gram-positive cocci, the genera *Abiotrophia*, *Peptostreptococcus*, *Streptococcus*, and *Stomatococcus* assume pivotal roles. Notably, some bacterial species, including *Streptococcus sanguinis* and *salivarius*, are known for producing antimicrobial compounds, making a significant contribution to the maintenance of oral health, and protecting dental structures from the damaging effects of enamel erosion. [25]. However, it is important to note that the full extent of each bacterium’s role in this complex interaction has not yet been fully elucidated.

Gram-positive rods, such as *Actinomyces*, *Bifidobacterium*, *Corynebacterium*, *Eubacterium*, *Lactobacillus*, *Propionibacterium*, *Pseudoramibacter*, and *Rothia*, contribute to the microbial consortium. Considering recent research, *Actinomyces naeslundii* has been discerned as proficient in utilizing urea as a nitrogen source, thereby strengthening its resistance against environmental acidification within the oral cavity. Consequently, it assumes a dominant role among dental plaque organisms, exerting a pronounced influence on the ecological processes of the plaque community [26]. Furthermore, *Lactobacillus*, *Bifidobacterium*, *Streptococcus*, and *Weissella* are all acknowledged as helpful probiotics in treating periodontal diseases, boosting oral health, and improving oral care as a whole [27].

In addition to the Gram-positive taxa, the Gram-negative cocci genera *Moraxella*, *Neisseria*, and *Veillonella* are prevalent inhabitants of the healthy oral cavity. Among the Gram-negative rods, the following genera are commonly observed: *Campylobacter*, *Capnocytophaga*, *Desulfobacter*, *Desulfovibrio*, *Eikenella*, *Fusobacterium*, *Hemophilus*, *Leptotrichia*, *Prevotella*, *Selemonas*, *Simonsiella*, *Treponema*, and *Wolinella* [28]. Many oral inhabitants are known for their ability to produce biofilms. *Fusobacterium nucleatum* exemplifies this capability through its unique role in mediating intergeneric coaggregation within oral biofilms. This involves its ability to bring diverse bacteria together, enabling them to create a well-organized community known as a biofilm. Furthermore, the involvement of *F. nucleatum* extends to the subsequent growth and development of these biofilms [29]. Within these biofilms, *Prevotella* organisms have a unique role. They are capable of regulating host residence and immune cell responses without requiring direct contact with the host. While *Prevotella* spp. are generally considered to be aerotolerant, their ability to thrive in aerated environments appears to be significantly influenced by their coaggregation with *F. nucleatum*. This coaggregation not only contributes to the structural maintenance of oral biofilms but also allows *Prevotella* spp. to benefit from the multispecies character of these biofilms [30]. The precise functions carried out by each bacterial genus offer a field for exploration and deeper study.

Fungi are estimated to constitute less than 0.1% of the total microbial population in the oral cavity [31]. The oral cavity is frequently associated with *Candida* spp., making it the most prevalent fungus observed. Additionally, other notable fungi identified in this environment include species belonging to the *Cladosporium*, *Aureobasidium*, *Saccharomycetales*, *Aspergillus*, *Fusarium*, and *Cryptococcus* groups [23]. When the host’s defense mechanism is weakened, fungi in the oral microbiome may still modulate immune responses or even develop into opportunistic infections [32]. Physical interactions between fungi and bacteria in the oral cavity influence the oral microbiota. *Candida* and other oral fungi act as bridging organisms, facilitating bacterial adhesion to their surfaces and epithelial cells. This leads to distinct sets of bacteria adhering to *Candida*-covered vs. non-covered cells. Additionally, these interactions contribute to bacterial resilience against antimicrobial agents, modulating the diversity of the oral microbiome [33]. Oral archaea exhibit significantly lower abundance and diversity compared to oral bacteria. In the oral cavity, all known archaeal species belong to the phylum *Euryarchaeota* and are characterized as methanogens, organisms that produce methane [2]. Studies have indicated that these archaea are found in higher quantities in individuals affected by periodontitis [34,35]. The composition of the oral microbiome is summarized in Table 1, showcasing the predominant phyla/genera of bacteria, fungi, and archaea, along with other notable genera.

In addition to the three primary domains of life—bacteria, fungi, and archaea—the oral microbiome also encompasses viruses. Oral microbiomes harbor extensive populations of viruses, predominantly bacteriophages (phages) that target specific bacterial species. These phages can exist either as free phage particles (virions) or as dormant prophages integrated within bacterial lysogens [36]. The phage population in dental regions, more indicative of periodontal disease and dental caries than the salivary phage population influenced by multiple oral surfaces, primarily includes *Caudovirus* families: *Siphoviridae* (generally lysogenic with intermediate host ranges), *Myoviridae* (typically lytic with broad host ranges), and *Podoviridae* (typically lytic with narrow host ranges). However, information on the periodontal phageome remains sparse [36,37,38].

Common eukaryotic viruses in the oral cavity of healthy adults include Herpesviridae, Papillomaviridae, Anelloviridae, and Redondoviridae [31,39]. Anelloviridae are the most common, with the newly discovered Redondoviridae as the second most common [31]. Prevalence measures depend on sampling methods, and amplification steps may favor small DNA circles, potentially boosting Anelloviridae and Redondoviridae detection. Redondoviridae prevalence ranges from 2 to 15% in various populations, and increased levels are observed in periodontitis patients and those with severe illnesses, though no evidence links them to disease causation [37,40]. Both viruses appear to be common commensals, detectable due to advancements in viral metagenomic sequencing [37]. Despite viruses being the dominant infectious agents, we have limited research on the complex viral communities in the human mouth, even though some now believe they could be commensals (23–25). However, it is critical to emphasize that while this hypothesis is gaining attention, substantial empirical evidence or explicit validation within the scientific literature is currently lacking to substantiate this assertion.

## 3. Factors Influencing the Oral Microbiome

### 3.1. Diet

Emerging research brings out the profound influence of dietary macronutrients and diet types on the composition and equilibrium of the oral microbiome. The presence of nutrients plays a crucial role, as seen in the notable impact of fermentable carbohydrates like sugars and starches, which strongly influence the proliferation of acid-producing bacteria, particularly *Streptococcus mutans*. These microbes drive the development of dental caries-generating acids that erode tooth enamel and initiate cavities [41]. Remarkably, the oral microbiome not only responds to dietary components, indicating its ability to change in response to the types of food consumed, but also actively participates in nutrient breakdown. This process involves specialized bacteria that degrade glycoproteins and sugars, a crucial step in nutrient utilization [42].

However, the impact of diet extends beyond nutrient breakdown. The influence of food choices on oral microbiome functions is shown by the fact that a nitrate-rich diet can promote the proliferation of nitrate-reducing bacteria. Dietary nitrate changes the salivary microbiome in people of various ages, with *Prevotella* and *Veillonella* populations declining and *Rothia* and *Neisseria* populations rising [43]. Notably, this modification in the salivary microbiota can affect the control of blood pressure and nitric oxide levels [44]. Furthermore, the complex relationship intertwines with immune responses and bacterial interactions. Antioxidants and anti-inflammatory compounds found in fruits and vegetables can modulate the immune response in the oral cavity, influencing the balance of various bacterial species [45].

### 3.2. Smoking

The influence of smoking on the oral microbiome is a multifaceted phenomenon with far-reaching implications for oral health. The relationship between smoking and the oral microbiome involves mechanisms such as the creation of anerobic conditions, compromised immune responses, altered salivary pH, and the antibacterial effects of cigarette smoke’s toxic compounds [46,47]. Nicotine, a substance found in tobacco leaves, has been shown to boost the virulence of oral microorganisms, possibly by stimulating the expression of virulence-related genes or promoting increased biofilm formation. Its influence on human cells is primarily through nicotine acetylcholine receptors (nAChRs) [48]. These alterations can disrupt the equilibrium within indigenous oral microbial communities, including beneficial species, while also creating a more favorable environment for the colonization of harmful microbes associated with oral diseases. Consequently, individuals who smoke may be at an increased risk of infections and oral health complications.

Tobacco smoke has a dual impact on the oral microbiome. First, it decreases oxygen levels, creating an environment ideal for anerobic bacteria that thrive without oxygen. Simultaneously, it increases free iron levels while inhibiting oral peroxidase activity, further promoting conditions favorable for anerobic bacterial growth. As a result of this, exposure to tobacco smoke prompts salivary cells to switch to anerobic glycolysis for energy production [49]. The alteration in the oral microbiota due to smoking is marked by a decrease in *Proteobacteria* and an increase in *Firmicutes* and *Actinobacteria*. These changes collectively contribute to the significant shifts observed in the composition of the oral microbiome in response to smoking [47]. Comprehensive studies utilizing advanced techniques are needed to illuminate the full scope of smoking’s impact, particularly on passive smokers, shedding light on potential interventions to counteract these effects and enhance oral health outcomes [23].

### 3.3. Alcohol Consumption

Excessive alcohol consumption can lead to an imbalance within the oral microbiome [50], causing potential repercussions for both oral and systemic health. Prolonged alcohol intake has been shown to impact various aspects of the oral environment, including the oral mucosa, salivary glands, and saliva composition [51]. Exposure to ethanol, whether acute or chronic, can result in significant changes in salivary function, characterized by the decreased flow rate and impaired secretion of crucial components like total protein and amylase [52,53]. Moreover, alcohol can impair the function of neutrophils, potentially leading to bacterial overgrowth and increased bacterial infiltration [54]. It can also reduce the production of inflammatory cytokines by monocytes, creating an environment conducive to microbial proliferation [55]. Additionally, ethanol’s adverse effects extend to dental health by promoting bone resorption and suppressing bone formation [56], impacting teeth, and supporting periodontal structures [57].

Notably, research has underscored a correlation between alcohol use and alterations in the oral microbiota, characterized by a diminished *Lactobacillales* abundance and an increased presence of *Neisseria*, *Streptococcus*, and *Prevotella* [58,59]. Interestingly, the oral microbiota has exhibited potential as an indicator of alcohol-induced liver damage [60]. Furthermore, diurnal changes within the oral microbiota, crucial for maintaining microbial balance, are reportedly disrupted by alcohol consumption, potentially contributing to disturbances within the microbial ecosystem and subsequent functional impairments [60]. Intriguingly, studies by Thomas et al. and Liao et al. have revealed distinct variations in the oral microbiota of alcohol consumers, including reduced bacterial richness and shifts in the prevalence of specific genera such as *Prevotella* and *Moryella*. Conversely, genera like *Lautropia*, *Haemophilus*, and *Porphyromonas* were found to be significantly diminished [61,62].

### 3.4. Other Factors That Influence the Oral Microbiome

In addition to diet, smoking, and alcohol consumption, several other factors significantly influence the composition and equilibrium of the oral microbiome. Poor oral hygiene practices, medical conditions, and medications play pivotal roles in shaping this complex ecosystem. Poor oral hygiene, characterized by inadequate brushing and flossing, promotes the accumulation of dental plaque—a biofilm predominantly composed of bacteria. This plaque buildup provides a conducive environment for the proliferation of pathogenic species, leading to an imbalance in the oral microbiome [63]. Moreover, various medical conditions exert substantial effects on oral microbial communities. Systemic diseases such as diabetes, CVD, neurological conditions, gastrointestinal diseases, and autoimmune disorders can disrupt immune responses and alter oral pH, creating favorable conditions for the overgrowth of pathogenic bacteria [64]. Furthermore, medications, notably antibiotics and immunosuppressants, can disturb microbial homeostasis by directly impacting bacterial populations or weakening host defenses [65]. Consequently, dysbiosis may arise, contributing to oral health issues. The use of broad-spectrum antibiotics can potentially disrupt the natural balance of beneficial oral microorganisms. In people without health issues, helpful microorganisms keep *Candida* in check by competing for nutrients and attaching to epithelial cells. Changes in these microorganisms might lead to the unchecked growth of *Candida*. Additionally, the broader effects of systemic antibiotic treatment could also contribute to oral candidiasis by affecting the immune system [66]. Moreover, the oral microbiome is a known site for the acquisition and transfer of resistance via horizontal gene transfer (HGT), which can contribute to the systematic development of antimicrobial-resistant infections [67]. HGT in the oral microbiome enables the rapid sharing of antibiotic resistance genes among microorganisms, leading to the emergence of antibiotic-resistant strains, reducing antibiotic efficacy and contributing to the overall problem of antibiotic resistance.

## 4. Oral Microbiome Dysbiosis

Oral microbiome dysbiosis refers to an imbalance or disruption in the composition and functioning of the oral microbial community [1]. Various factors, such as poor oral hygiene, dietary habits, medical conditions, medications, and lifestyle choices, can lead to this disturbance. This imbalance disrupts the previously harmonious interactions between different bacteria, altering the overall microbial ecology. The repercussions of oral microbiome dysbiosis are significant, contributing to common oral health issues like dental caries (cavities), periodontal diseases (gum infections), halitosis (bad breath), and oral candidiasis (oral thrush). Emerging research also indicates potential links between oral microbiome dysbiosis and systemic health conditions such as cardiovascular disease (CVD), diabetes, and cognitive disorders [8]. A key characteristic of oral microbiome dysbiosis is the overgrowth of harmful bacteria and a decline in beneficial microorganisms [68]. This imbalance fosters an environment where pathogens can thrive, causing inflammation and damage to oral tissues. Moreover, dysbiosis can disrupt the natural processes that maintain and repair tissues, further aggravating oral health problems. 

## 5. Oral–Gut Axis

The oral and gut microbiomes play crucial roles in human health, each harboring a distinct microbial community. These communities are typically separated by barriers such as the oral–gut barrier, physical distance, and chemical obstacles like gastric acid and bile [69]. However, recent research has shed light on the potential microbial translocation between these two ecosystems, highlighting the interconnected nature of the oral–gut microbiome axis. Several factors can influence the translocation of oral microbiota to the gut. Individuals at the extremes of life, such as neonates and the elderly, may have fewer effective barriers, allowing for increased microbial movement [70,71]. Certain bacteria, like *Bifidobacterium*, have been detected in the neonatal gut and oral fluids [72]. Additionally, decreased gastric acidity can shift the gut microbiota composition to resemble the oral microbiome. This suggests that under certain conditions, oral microbes can bypass barriers and successfully translocate to the gut [73]. Microbial translocation is not limited to physiological states. Pathological conditions, such as IBD and colon cancer, have shown evidence of oral bacteria colonizing the gut mucosa [74]. This suggests that disruptions in mucosal homeostasis can lead to the invasion of oral commensals into the gut, potentially contributing to disease progression.

Two main hypotheses have been proposed for the transmission of oral bacteria to the gut: the hematogenous route and the enteral route (Figure 1). The hematogenous route suggests that oral bacteria enter the bloodstream, circulate, and colonize the gut mucosa. Meanwhile, the enteral route proposes that bacteria from the mouth travel through the stomach to the intestines. Although the human body possesses defenses against microbial invasion, instances can arise where these barriers are compromised, allowing for bacterial movement [75]. The findings of oral-to-gut microbial translocation implicate human health. Antibiotic use and conditions like achlorhydria (absence of hydrochloric acid in the gastric secretions) can diminish barriers, facilitating the colonization of oral bacteria in the gut [76,77]. Certain pathogens, like *Klebsiella* spp. and *Porphyromonas gingivalis*, possess antibiotic resistance and acid-resistant properties, respectively, enhancing their ability to translocate to the gut [76,78]. The occurrence of oral commensal bacteria in the gut is an uncommon and abnormal event, resulting from a process known as ectopic colonization, where these bacteria are found in a location where they do not typically belong. This presence of oral commensals in the gut is considered a characteristic feature of various diseases, including IBD [79]. Several known oral bacterial species belonging to genera such as *Staphylococcus*, *Porphyromonas*, *Veillonella*, *Fusobacterium*, *Actinomyces*, and *Parvimonas* have been detected within the gut of patients with gastrointestinal disorders [77]. The emerging understanding of oral–gut microbial translocation deepens the complex interplay between these two microbial ecosystems.

## 6. Oral Microbiome and Oral Diseases

### 6.1. Dental Caries

Dental caries stand as the most prevalent infectious disease within the oral environment [80] primarily driven by bacterial pathogens and characterized by progressive dental hard tissue deterioration [81]. Recent investigations have unveiled the pivotal role of the oral microbiome in caries onset and progression. *Prevotella* spp., *Lactobacillus* spp., *Dialister* spp., and *Filifactor* spp. have surfaced as potential contributors to caries pathogenesis and advancement [82]. Caries development is attributed to the acidic by-products of sugar metabolism by acidogenic bacteria, inducing demineralization and cavitation [83]. All these factors, such as ecological imbalances, trends towards acidogenic environments, and the acidification of biofilms, contribute to the emergence of caries [84]. The caries biofilm, a complex and highly active ecosystem, initiates the accumulation of dental plaque on the tooth’s surface. Within this environment, various microorganisms colonize, with *S. mitis* and *S. mutans* being among the initial colonizers [85,86].

The formation of the biofilm begins when a salivary glycoprotein film (dental pellicle) coats a tooth surface [87]. These initial colonizers create extracellular polymers (EPSs) that enhance the adherence of other microorganisms to the biofilm. EPS-rich environments provide ideal binding sites for acid-producing bacteria. Acid-producing bacterial species, such as *Veillonella*, *Lactobacillus*, and *Propionibacterium*, become part of the dental biofilm. They contribute to cariogenic conditions in the mouth by producing acids [88]. *S. mutans* has cariogenic potential; excels in acid production, allowing it to metabolize a wide range of carbohydrates into organic acids; and exhibits remarkable acid resistance, enabling it to thrive even in low-pH environments. Additionally, it can synthesize EPSs that not only promote its growth but also offer protection to its cells in challenging and harsh conditions. These attributes collectively underscore the significant role of *S. mutans* in the pathogenesis of dental caries [89]. *S. mutans* utilizes a crucial virulence factor, specifically three enzymes known as glucosyltransferases (GtfBCD), to initiate and establish cariogenic biofilms [90]. Bacterial diversity within caries-associated microbiota is modulated by competitiveness, as evidenced by varying compositions across different carious sites [90,91]. *S. mutans* [92], long considered a key player, has been joined by acidogenic species like *Bifidobacterium* [93] and *Lactobacillus* [94]. The multifaceted cariogenic consortia include *Prevotella*, *Corynebacterium*, *Veillonella*, and *Capnocytophaga*, collectively contributing to local pH reduction and tissue demineralization [93,95].

### 6.2. Gingivitis

Gingivitis is described as inflammation of the gums, typically caused by the accumulation of microbial plaque, or bacteria, on the surface of the teeth due to ineffective tooth brushing. Thus, the importance of effective tooth brushing cannot be overstated, as it is essential for removing food debris and preventing the further growth of plaque [96]. Bacteria present in accumulated plaque on the tooth surface can penetrate the gingival tissue, particularly the gingival sulcus, making the marginal area prone to microbial infection. Common microbial species associated with gingivitis include *P. gingivalis*, *A. actinomycetemcomitans*, *Streptococcus* sp., *Fusobacterium* sp., *Actinomyces* sp., *Veillonella* sp., *Treponema denticola*, and *Prevotella intermedia* [97]. If untreated, gingivitis has the potential to advance to periodontitis, a condition that can lead to irreversible damage not only to the gums but also to the surrounding bone that supports the teeth [96].

### 6.3. Periodontitis

The oral microbiome plays a pivotal role in the development and progression of periodontal diseases, encompassing gingival diseases and periodontitis [98]. Chronic periodontitis, a prevalent form of periodontal disease, spans a wide age range and evolves from gingivitis to deeply affecting periodontal tissues. Dental plaque bacteria emerge as the principal culprits in the onset of periodontal disease [99]. Plaque exists in both supragingival and subgingival areas, fostering microbial biofilms that thrive on tooth surfaces. The crevicular epithelium and gingival crevice act as pivotal habitats for microbial initiation, supported by gingival crevicular fluid and a favorable anerobic environment [10]. The subgingival plaque, comprising more than 500 species, undergoes compositional shifts during disease progression [100]. In healthy conditions, Gram-positive cocci and rods dominate as early colonizers, while *F. nucleatum* bridges bacterial coaggregation [100]. However, as gingivitis advances, Gram-negative bacteria like *Prevotella* spp. and *F. nucleatum ss. polymorphum* surge, associated with elevated inflammatory cytokines such as IL-1, IL-6, IL-17, TNF-α, and IL-23 [98,101,102].

IL-17 plays a stimulatory role in activating T helper (Th)-17 and B cells, which, when activated, elevate the expression of receptor activator of nuclear factor kappa-B ligand (RANKL). RANKL is also directly activated by the neutrophils that have migrated to the site. RANKL, in turn, triggers the activation and maturation of osteoclast precursors into active osteoclasts, which are responsible for bone resorption. Simultaneously, the recruited neutrophils contribute to tissue degradation by inducing the production of enzymes like matrix metalloproteinases (MMPs) and cytotoxic substances, including reactive oxygen species (ROS). These interactions among microbes, the innate immune response, and the adaptive immune response illustrate key mechanisms involved in the persistence of inflammation, which, if left unresolved, leads to tissue destruction [103]. The red complex bacteria—*P. gingivalis*, *Treponema denticola*, and *Tannerella forsythia* [24]—emerge as keystone pathogens, triggering an imbalance in host–bacterial interactions and resulting in bone loss [104,105]. Furthermore, recent investigations highlight the systemic implications of periodontal diseases. Periodontitis not only poses local health risks but also acts as a potential risk factor for systemic conditions like CVD, diabetes, and cancer [9].

### 6.4. Halitosis

Halitosis, commonly referred to as oral malodor or bad breath, is a frequent oral condition marked by an unpleasing or disagreeable odor that originates within an individual’s mouth [106]. This widespread condition impacts a significant portion of the worldwide population, ranging from 15% to 60%, and is characterized by the emission of an offensive odor from the mouth, nasal passages, and throat regions [107]. Intra-oral halitosis (IOH), closely tied to microbial interactions within the oral environment, is a prominent facet of this condition. Anerobic bacteria give rise to volatile compounds, with key volatile sulfur compounds (VSCs) including hydrogen sulfide, dimethyl sulfide, dimethyl disulfide, and methyl mercaptan [107,108]. Oral bacteria, such as *Prevotella* spp. [109], *Bacteroides* spp., and *Actinomyces* spp. play vital roles in halitosis by producing VSCs during the biodegradation of sulfur-containing amino acids [108,109]. These sulfur-containing amino acids, such as L-cysteine and L-methionine, originate from the saliva, the blood, the oral mucosa, and epithelial cell debris. The enzymatic reactions involving L-cysteine and L-methionine are crucial in sulfur-containing amino acid metabolism. L-cysteine undergoes desulfhydration, catalyzed by the L-cysteine desulfhydrase enzyme, resulting in the formation of pyruvate, ammonia, and hydrogen sulfide. On the other hand, L-methionine is acted upon by the L-methionine α-deamino-γ-mercaptomethane-lyase enzyme, leading to the generation of α-ketobutyrate, ammonia, and methyl mercaptan [110]. In addition, VSCs, particularly hydrogen sulfide and methyl mercaptan, exhibit significant toxicity. These observations imply that VSCs not only play a role in causing bad breath but could also be involved in the development of periodontitis [111].

Gram-negative bacteria, including *Prevotella* (Bacteroides) *melaninogenica*, *Treponema denticola*, *Porphyromonas gingivalis*, *Porphyromonas endodontalis*, and others like *Enterobacteriaceae*, *Tannerella forsythia*, *Eikenella corrodens*, and various *Fusobacterium* species, are implicated as likely contributors to oral malodor. However, the lack of a clear association between a specific bacterial infection and halitosis suggests that malodor is the result of intricate interactions among multiple oral bacterial species [112].

Earlier studies by Persson and collaborators highlighted the production of H_2_S by bacteria such as *Treponema denticola* and *Bacteroides intermedius*, alongside *Peptostreptococcus*, *Eubacterium*, and *Fusobacterium* [113]. Additionally, investigations by other researchers revealed the dominance of *Veillonella* spp., *Actinomyces* spp., and *Prevotella* spp. in patients experiencing oral malodor [114]. Conversely, *Neisseria* spp., *Fusobacterium* spp., and *Porphyromonas* spp. were noted to be most prevalent in individuals with elevated H_2_S concentrations, while *Veillonella* spp., *Prevotella* spp., *Megasphaera*, and *Selenomonas* predominated in those exhaling primarily CH_3_SH [115]. Notably, lower levels of *Streptococcus* and *Granulicatella* were evident, while higher proportions of *Leptotrichia*, *Peptostreptococcus*, *Eubacterium*, and *Fusobacterium* were detected across both (H_2_S and CH_3_SH) malodor groups [115].

### 6.5. Taste Impairment

Taste ability is notably compromised in acutely hospitalized elderly individuals who exhibit decayed teeth, elevated levels of oral bacteria linked to caries, poor oral hygiene, and dry mouth [116]. It has been proposed that the loss of taste associated with poor oral health may be attributed to toxins and inflammatory products produced by oral bacteria [117]. A study revealed that patients with high *Lactobacilli* growth experienced particularly impaired sour taste perception [116]. *Lactobacilli* thrive in acidic environments and produce acid themselves, potentially causing an adaptation in sour taste perception and increasing the taste threshold for sour flavors. Additionally, we found that inadequate oral hygiene correlated with a reduced total taste score and diminished perception of salty taste [118,119]. Saliva plays a crucial role in transporting food particles and taste stimuli to the taste buds in the oral cavity, and reduced saliva flow has been linked to taste impairment [120]. Hospitalized elderly individuals frequently consume a high number of medications, which can induce xerostomia (dry mouth) and hypo-salivation [121], resulting in the loss of taste. Findings by [122] and [123] reveal reduced taste sensitivity in individuals with Sjøgren’s syndrome, characterized by oral dryness, compared to controls.

### 6.6. Burning Mouth Syndrome

Burning mouth syndrome (BMS) is a condition characterized by a persistent burning sensation in the mouth without any apparent lesions [124]. Recent research suggests a potential association between BMS and the oral microbiota, shedding light on the role of oral bacteria in the etiology of this syndrome. A study investigating the oral microbiota signature of BMS patients found marked differences in microbial profiles between BMS patients and healthy controls [125]. The composition of bacterial strains, including dominant genera, was distinct in BMS patients compared to controls. *Streptococcus*, *Rothia*, *Bergeyella*, and *Granulicatella* were prevalent in the BMS group, while *Prevotella*, *Haemophilus*, *Fusobacterium*, *Campylobacter*, and *Allorevotella* were more common in healthy controls [125]. This correlation between the oral microbiome composition and BMS suggests a potential role of oral bacteria in the pathogenesis of BMS.

### 6.7. Oral Thrush

Oral thrush, also known as oral candidiasis, is a fungal infection that affects the oral mucosa. It is caused by an overgrowth of *Candida albicans* [126]. *Candida krusei*, *Candida stellatoidea*, *Candida tropicalis*, *Candida glabrata*, *Candida guilliermondii*, and *Candida dubliniensis* are other species seen in oral candidiasis lesions [127]. The symptoms of oral thrush can include white patches on the tongue, inner cheeks, and roof of the mouth, as well as redness and soreness in the affected areas. In severe cases, it can cause difficulty swallowing and a burning sensation in the mouth [128]. Factors such as poor oral hygiene, a weakened immune system, or certain medical conditions like diabetes or HIV/AIDS can increase the risk of oral thrush [129]. These factors can create an imbalance in the natural flora of the mouth, allowing the *Candida* fungus to overgrow and cause an infection. Several processes contribute to *Candida albicans* pathogenicity (Figure 2). Initially, it adheres to host surfaces through weak and reversible interactions that are influenced by both hydrophobic and electrostatic forces [130]. The process of adhesion is facilitated through the presence of specific host tissue receptors, notably the glycoproteins ALS (agglutinin-like sequence) [131] and HWP1 (hyphal wall protein) (100). *C. albicans* can adhere to epithelial cells using various host cell receptors, such as EphA2 (through β-glucan) and E-cadherin (through ALS3).

Additionally, host cell transglutaminases create direct links between *C. albicans* and the epithelial surface by interacting with HWP1. Among the family of eight glycosylphosphatidylinositol (GPI)-linked Als proteins, ALS3 stands out as a pivotal player in mediating the attachment of fungal hyphae to epithelial cells and facilitating their subsequent penetration into host cells [132,133]. HWP1 exhibits elevated expression levels during oral infections in humans [134], and it serves as a target for epithelial transglutaminases, facilitating robust and covalent connections with various epithelial proteins [135]. The interaction with oral bacteria can promote biofilm formation. *Candida* spp. readily forms biofilms, microbial communities encased in an extracellular matrix. Biofilms contribute to increased virulence and resistance to antimicrobial agents. In oral candidiasis, biofilms on dentures are a major factor [136]. For instance, *Streptococcus gordonii* is a common resident that interacts with *C. albicans* in the oral cavity, promoting the formation of hyphal structures and biofilms. This interaction is facilitated by the direct binding of ALS3 on *C. albicans* to the surface protein SspB (cell-wall-associated protein) on *S. gordonii*, highlighting the complex interactions between microbes in the oral environment that impact biofilm formation and colonization [137]. *C. albicans* can change into an invasive filamentous form after adhering to host surfaces, which significantly improves its ability to penetrate epithelial tissue [138,139].

*Candida* can phenotypically switch between its yeast and hyphal states. *C. albicans* worsens host tissue damage by releasing enzymes outside the cell. *Candida* secretes enzymes like secreted aspartyl proteinases (SAPs), phospholipases, and lipases. These enzymes can degrade host immune factors, such as antibodies and antimicrobial peptides, reducing the effectiveness of host defenses [140]. For tissue invasion, the shift to the hyphal form is crucial [141]. Hyphae can infiltrate and harm the epithelial cells, resulting in inflammation and distinctive white patches that appear in oral thrush. Thigmotropism, a directional growth response to surface contact, is a characteristic of hyphal cells and aids in their ability to successfully explore and infiltrate host tissues [142]. *Candida* can release candidalysin, a hypha-specific toxin that promotes immunological activation and tissue destruction [143]. Candidalysin, when acting upon oral epithelial cells, triggers the entry of calcium ions and the release of lactate dehydrogenase (LDH), both of which are indicative of cellular harm and the destabilization of the cell membrane [144]. In addition to this, *Candida* engages in a complementary, passive process called endocytosis. ALS3 plays a dual role as both an adhesin and an invasin, and it collaborates with Ssa1, a heat shock protein. Together, they facilitate the endocytosis of *C. albicans* into epithelial cells through their interaction with E-cadherin. In this process, fungal enzymes and invasins on the hyphae bind to and degrade inter-epithelial cell junctional proteins like E-cadherin, enabling the yeast to penetrate and enter host epithelial cells [145].

## 7. Oral Microbiome and Systemic Diseases

### 7.1. Gastrointestinal Disorder: Inflammatory Bowel Disease

Inflammatory bowel disease (IBD), a chronic inflammatory condition affecting the gut, is known to be associated with oral microbial dysbiosis [74]. In healthy conditions, the mucosal barrier largely prevents microbial translocation from the oral cavity to the gut [146]. However, IBD’s altered gut epithelial permeability allows oral-resident bacterial strains to infiltrate the gut microbiome [147,148]. *F. nucleatum*, typically an oral inhabitant, has been detected in the intestines of IBD patients, underscoring the potential oral–gut axis in disease pathogenesis [149]. This notion is supported by in vivo studies wherein oral microbiota transplantation led to gut dysbiosis and exacerbated visceral hypersensitivity, further accentuating the role of oral microbes in influencing gut health [76,150]. Periodontitis exhibits a strong association with IBD. Keystone pathogen *P. gingivalis* disrupts intestinal barrier function, impacting the gut microbiome composition and inciting inflammation [101,104]. *P. gingivalis* enters the gastrointestinal tract and triggers an inflammatory response in the ileum. This inflammation is a localized immune reaction intended to combat the invading pathogen. During the inflammatory response, there is an activation and proliferation of a particular subset of immune cells known as IL9+ CD4+ lamina propria T cells. These T cells are known for their production of interleukin-9 (IL-9), a cytokine that plays a role in immune responses [151] (Figure 1).

Moreover, pediatric Crohn’s Disease (CD) patients, a type of IBD, exhibited an enrichment in resident oral bacteria, including *Enterobacteriaceae*, in the gut [76]. *Klebsiella* spp. derived from the oral cavity have been identified as potent inducers of Th1-mediated inflammation in the gut. This implies a potential contribution of oral bacteria to CD pathogenesis [77]. Collectively, the oral microbiome manifests in IBD’s complex etiology. The oral–gut axis suggests that oral dysbiosis can influence IBD pathogenesis, while alterations in gut health reciprocally impact the oral microbiome [74]. It is promising that further research into the precise mechanisms and microbes responsible for these interactions will lead to new strategies for treating IBD.

### 7.2. Cardiovascular Diseases: Atherosclerosis

Studies have revealed the connection between oral dysbiosis and cardiovascular disease (CVD), as well as the impact of CVD on the oral microbiota. In the context of periodontal diseases, an increase in Gram-negative bacterial populations, such as *P. gingivalis*, *T. denticola*, and *T. forsythia*, can exacerbate CVD by triggering local inflammation, which in turn stimulates systemic inflammatory responses, oxidative stress, immune activation, and platelet aggregation factors associated with CVD [152,153]. Pathogens such as *S. mutans* [154] and *P. gingivalis* [155] are increased in patients with both periodontal disease and atherosclerosis. Emerging evidence suggests that the relationship between oral microbiota and CVD is bidirectional, with CVD impacting the oral microbiota through systemic inflammation and immune activation. Microbial invasion of the bloodstream [156], facilitated by oral–gut microbiota transfer [79,157] and bacteremia, may exacerbate systemic inflammation, leading to a cascade of effects that further disrupt the oral microbial balance.

Metabolites produced by the oral microbiota are gaining recognition for their potential to influence cardiometabolic health. Elevated concentrations of bacterial surface molecules like lipopolysaccharides (LPSs) and bacterial flagellins stimulate the production of pro-inflammatory mediators and cytokines [158,159]. These molecules trigger inflammatory pathways such as the nuclear factor kappa-B (NF-κB) and matrix metalloproteinase 9 (MMP9) pathways, contributing to inflammation and immunoreactivity [155]. *Aggregatibacter actinomycetemcomitans*, linked to chronic periodontitis, releases leukotoxin A (LtxA), which kills white blood cells by causing cofilin dephosphorylation and actin depolymerization [160]. LtxA disrupts enzyme function in neutrophils, causing hypercitrullination and releasing altered proteins that could trigger autoimmune responses, possibly contributing to atherosclerosis [161]. Several additional bacterial species have also been implicated in the interaction between disrupted oral microbial balance and CVD. These species, including *Prevotella intermedia*, *Prevotella nigrescens*, *Campylobacter rectus*, *Parvimonas micra*, *P. endodontalis*, *Eubacterium timidum*, *Eubacterium brachy*, and *Eubacterium saphenum*, have been linked to both oral dysbiosis and the development of CVD [162,163,164].

Apart from this, autoimmune processes constitute a crucial mechanism in the development of atherosclerosis. One intriguing aspect of this immune response involves heat shock proteins (HSPs), which are part of a family of molecular chaperones that protect cells during stress [165]. These proteins are present in the host as well as in bacteria, including *P. gingivalis*. In the case of *P. gingivalis*, its HSP60 equivalent, known as GroEL, can trigger both a humoral and cellular immune response in humans [166]. This immune response can lead to chronic activation, as antibodies originally primed for bacterial HSPs may cross-react with human HSP60 due to structural similarities. This chronic immune activation fuels inflammation within arterial walls, promoting the accumulation of plaque composed of cholesterol, immune cells, and debris. Over time, the plaque buildup narrows arteries, reducing blood flow and raising the risk of cardiovascular events like heart attacks and strokes when unstable plaques rupture [161].

### 7.3. Endocrine Disorders: Diabetes Mellitus

Diabetes mellitus, marked by hyperglycemia and inflammation coupled with elevated oxidative stress, creates an environment conducive to microbial dysbiosis. Subsequently, alterations within the oral microbiome have been reported, with distinct bacterial shifts observed in diabetic subjects [167]. Elevated proportions of *Capnocytophaga* [168], *P. gingivalis*, *T. forsythia* [169,170], and others have been noted, hinting at potential microbial contributors to diabetic-associated oral complications. However, the findings remain diverse and inconsistent, partly due to limited bacterial profiling and confounding factors. A newly gained understanding of the underlying mechanisms has revealed how diabetes-induced changes in host responses play a significant role [171]. Individuals with diabetes experience elevated inflammatory reactions, impaired neutrophil activity, and a diminished ability to clear bacteria. This cascade of events contributes to exacerbated periodontal inflammation and tissue damage. Moreover, the dysregulated cytokine environment, including tumor necrosis factor (TNF), fuels inflammation, exacerbating periodontal degradation [172,173]. Oral microbes in diabetic individuals exhibit high virulence, potentially fueling disease progression. Diabetes-induced IL-17 elevation further amplifies oral microbial pathogenicity, intensifying the risk of periodontal deterioration [167]. Insightfully, interventions targeting IL-17 have shown promise in mitigating the pathogenic potential of the oral microbiome in diabetes.

### 7.4. Obesity

Obesity is characterized by chronic low-grade inflammation, recognized as a key link between obesity and metabolic dysfunction [174,175]. This interaction, termed “meta inflammation”, is particularly notable in adipose tissue [176]. Recent research indicates a connection between obesity and the oral microbiome, suggesting oral bacteria might play a role in obesity’s development and progression [177]. Distinct shifts in the composition of the oral microbiome have been observed in individuals with obesity compared to non-obese individuals [178]. Intriguingly, lean (non-obese) individuals showcase elevated populations of *Firmicutes*, *Bacteroidetes*, and *Spirochaetes* [179,180] in sublingual plaque, while their obese counterparts exhibit increased levels of *Proteobacteria*, *Chloroflexi*, and *Firmicutes* [181,182]. Alterations in bacterial populations have been associated with changes in metabolic homeostasis. An area of investigation focuses on the migration of oral bacteria to the gut, subsequently influencing the composition of the gut microbiome. This “oral–gut-axis” hypothesis suggests that oral microbes can influence distant organs, potentially contributing to systemic inflammation and metabolic perturbations. Additionally, the concept of an “oral–blood axis” highlights the potential for oral bacteria and inflammatory molecules to enter the bloodstream, further propagating inflammation and metabolic dysfunction [175]. Moreover, studies have identified a significant link between periodontal disease, oral dysbiosis, and obesity-associated inflammation [183]. Intriguingly, overweight individuals exhibit altered salivary bacterial composition, possibly serving as an early indicator of developing obesity.

### 7.5. Neurological Disorders: Alzheimer’s Disease

Alzheimer’s disease (AD) is a progressive and neurodegenerative disorder predominantly affecting individuals aged over 65 years, characterized by significant cognitive decline that impairs daily activities [184]. Recent research has unveiled a robust connection between AD and the composition of the oral microbiota. Notably, specific pathogens within this microbial array, namely *P. gingivalis*, *T. forsythia*, and *T. denticola*, play central roles in periodontitis, inciting a cascade of inflammatory responses that have been implicated in AD progression [185,186,187,188]. The oral microbiota’s production of inflammatory agents introduces a novel perspective on systemic infection and inflammation. These mediators can traverse the bloodstream, influencing remote tissues and organs, including the brain. This suggests that the oral microbiota may act as a potential primary agent for AD, provoking neuroinflammation and contributing to the disease’s etiology [189]. Interestingly, *P. gingivalis*, an oral pathogen, has been detected in brain tissue [184]. It can directly impact AD through various mechanisms. These microbes gain access to the brain via routes such as transient bacteremia, root canal pathways, or gingival crevices, where they provoke direct damage to the central nervous system (CNS), initiating an immune cascade characterized by microglial activation and inflammatory responses [190]. This process may induce the production of Aβ protein, which, while normally protective, could be overproduced in response to oral microbes, potentially contributing to the formation of Aβ plaques in AD [191].

Furthermore, the blood–brain barrier (BBB) may become more permeable in elderly AD patients, aiding oral microbe entry. Some oral microorganisms and their secreted substances can even directly alter BBB permeability. Indirectly, chronic oral diseases weaken the oral mucosal barrier, permitting cytotoxins and pro-inflammatory factors to enter the bloodstream and trigger systemic inflammation. These factors can then reach the CNS, indirectly influencing AD by inciting neuroinflammation and cognitive impairment associated with the disease [192]. While Aβ peptides have demonstrated antimicrobial efficacy against a range of bacteria and *Candida albicans*, their prolonged accumulation due to persistent colonization by pathogenic oral bacteria might lead to the formation of amyloid plaques and tau protein hyperphosphorylation, ultimately culminating in brain tissue damage [190,193].

### 7.6. Parkinson’s Disease

Parkinson’s Disease (PD), a prevalent neurodegenerative disorder primarily recognized for its impact on movement control, poses significant challenges to the aging population [194,195]. It is a disorder that affects memory. PD’s core manifestation lies in compromised fine and gross motor skills. The onset of PD, frequently late in life, may not exclusively stem from genetic inheritance but rather a combination of multifactorial contributors, including genetic predisposition, environmental risks, and aging [196]. Recently, many studies have shown a correlation between *P. gingivalis* infection and PD [197,198,199,200]. Studies have revealed that gingipain R1 (RgpA), a virulence factor produced by *P. gingivalis*, can be detected in the bloodstream, indicating systemic dissemination of the bacterium [197]. Additionally, PD patients exhibit a hypercoagulable state characterized by hyperactivated platelets and fibrinogen amyloid features, which may be influenced by *P. gingivalis* and its products [197,201]. Furthermore, *P. gingivalis* is implicated in inducing systemic inflammation, as evidenced by the presence of its lipopolysaccharides (LPSs) and gingipain in circulation [201]. This systemic inflammation is known to play a role in the pathogenesis of PD. Animal studies have provided further insights into the potential mechanisms linking *P. gingivalis* to PD. The oral administration of live *P. gingivalis* to mice with a genetic mutation associated with PD leads to neurodegeneration in the substantia nigra, the activation of microglia, and the upregulation of IL-17A secretion and its receptor IL-17RA [199]. These findings suggest that *P. gingivalis* may contribute to PD pathogenesis by inducing systemic inflammation, promoting hypercoagulability, and exacerbating neurodegeneration. However, further research is needed to fully elucidate the complex interplay between the oral microbiota, systemic inflammation, and neurodegenerative diseases like PD.

### 7.7. Autoimmune Conditions: Rheumatoid Arthritis

Rheumatoid arthritis (RA), a chronic autoimmune disease characterized by symmetrical inflammation and joint pain, has a complex etiology involving genetic and environmental factors [202]. Recent research has unveiled a compelling connection between the oral microbiota and RA pathogenesis, shedding light on the role of specific microorganisms and their molecular mechanisms in triggering and exacerbating the disease. Among the key players, *P. gingivalis* stands out as a significant pathogen implicated in both periodontitis and RA [203]. *P. gingivalis* is equipped with virulent factors, including the bacterial protein arginine deiminase (PPAD), capable of citrullinating host proteins [204]. This citrullination has been postulated to initiate an autoimmune response, culminating in the production of anticitrullinated protein antibodies (ACPAs) that are characteristic of RA. Neutrophil extracellular traps (NETs) generated during infection further contribute to the citrullination of proteins catalyzed by PAD4 enzymes and ACPA production [205,206]. The presence of ACPAs has been correlated with *P. gingivalis* exposure and has been proposed as a crucial link between periodontal inflammation and RA onset (Figure 3).

Beyond *P. gingivalis*, a broader dysbiosis of the oral microbiota has appeared as a potential factor in RA development. Elevated levels of *Prevotella*, *Veillonella*, and anerobic species, such as *Lactobacillus salivarius*, have been observed in RA patients [207]. Conversely, decreased levels of beneficial microbes, such as *Haemophilus* spp., have been noted. This dysbiosis can lead to altered immune responses, including the overactivation of T helper 17 (Th17) cells, which play a pivotal role in autoimmune processes. Th17 cells produce pro-inflammatory cytokines like IL-17 [208]. Elevated levels of IL-17 have been found in the synovium of RA patients and are associated with disease severity [209]. Moreover, the interplay between microbial communities and immune cells can exacerbate joint inflammation and bone destruction in RA [210]. Recent investigations have also implicated *A. actinomycetemcomitans*, a periodontal bacterium, in RA pathogenesis. Leukotoxin A (LtxA) secreted by *A. actinomycetemcomitans* triggers neutrophil hypercitrullination and the subsequent release of citrullinated autoantigens, further fueling the ACPA response [211]. Some oral microbiome molecules structurally resemble human self-antigens, potentially leading to cross-reactivity with host proteins. For instance, *P. gingivalis’* α-enolase protein shares homology with human α-enolase, an autoantigen in RA. Antibodies against citrullinated α-enolase (anti-CEP-1) have been found in individuals with chronic PD, suggesting a connection [212].

### 7.8. Systemic Lupus Erythematosus

Systemic Lupus Erythematosus (SLE) is a complex autoimmune disease that impacts various organs and tissues, disproportionately affecting women of reproductive age [207]. Although its etiology remains elusive, recent research has shed light on the potential involvement of the oral microbiome in SLE pathogenesis. Numerous studies have demonstrated alterations in the oral microbiota composition in individuals with SLE. This microbial dysbiosis extends beyond the oral cavity, possibly contributing to the systemic inflammation characteristic of SLE [213]. A study conducted by Pessoa et al. made a significant observation regarding the oral microbiota in individuals with SLE compared to healthy controls. Fourteen distinct subgingival bacteria in SLE patients were enriched, suggesting a distinct microbial profile associated with the disease. Interestingly, this elevated microbial abundance was particularly evident in certain bacteria, specifically *T. denticola* and *T. forsythia* [214]. Moreover, an increased diversity of oral microbiota was observed in SLE patients’ saliva, with *Veillonella*, *Streptococcus*, and *Prevotella* as the dominant species [215].

Periodontitis may exacerbate SLE through activation of the Toll-like receptors (TLRs), such as TLR2 and TLR4. These receptors are pivotal in recognizing microbial components and initiating immune responses [216]. The interesting concept of molecular mimicry arises in the context of SLE. Cross-recognition between self-antigens, like Ro60, and homologous peptides from oral bacteria suggests a potential role for bacterial antigens in triggering autoimmune responses. These observations highlight the oral microbiota’s capacity to induce and perpetuate aberrant immune reactions in SLE [216]. Elevated levels of pro-inflammatory cytokines, such as interleukin-6 (IL-6), IL-17A, and IL-33, have been detected in the saliva of SLE patients with chronic periodontitis. These cytokines play a pivotal role in promoting immune cell activation and differentiation, thereby intensifying autoimmune responses and tissue damage [217,218,219]. The complicated relationship between oral microbial dysbiosis, periodontitis-induced inflammation, and systemic immune dysregulation emphasizes the potential impact of the oral microbiota on SLE progression.

### 7.9. Cancer

Within the realm of cancer research, specific bacterial strains have emerged as influential contributors to cancer progression, shedding light on the intricate connections between the oral microbiome and this complex disease. Notably, two bacterial strains, particularly *F. nucleatum* and *P. gingivalis*, have been recognized as agents that contribute to cancer progression. *F. nucleatum* employs a range of surface proteins to infiltrate diverse cell types, potentially initiating tumor development in the mouth, colon, and esophagus. Similarly, *P. gingivalis* is connected to periodontal harm and can support tumor formation by invading neighboring tissues. This bacterium can also travel to other sections of the oral and digestive systems, and its association with pancreatic cancer has been observed [220]. Certain bacteria produce specific substances with potent cancer-causing properties. For instance, lipopolysaccharide, which is generated by numerous anerobic oral microorganisms, including *S. oralis*, *S. mitis*, *S. sanguinis*, *Lactobacillus fermentum*, *Lactobacillus acidophilus*, and *Bifidobacterium adolescentis*, can induce inflammatory mechanisms linked to the development of cancer [221].

A comprehensive overview of the associations between the oral microbiome and various diseases is mentioned in Table 2. It is noteworthy that the organisms discussed are referred to as commensal when they are present at low levels, typical of a healthy microbiome.

## 8. Oral Health and Dental Care Practices

### 8.1. Probiotics

Dental probiotics contribute to the balance of the oral microbiota by colonizing the oral mucosa, producing specialized metabolites to maintain the host microbiota’s ecological equilibrium, and promoting a healthy level of immunity [222]. Scientifically studied probiotic strains such as *Streptococcus salivarius M18*, *Streptococcus salivarius K12*, *Lactobacillus plantarum*, *Bifidobacterium lactis*, *Lactobacillus reuteri*, and *Lactobacillus salivarius* have demonstrated their efficacy in supporting oral health. These strains are known to combat the bacteria responsible for bad breath and gum inflammation. *S. salivarius* can generate powerful antibacterial proteins known as bacteriocins, which can be harmful for the growth of various bacterial species [223]. Furthermore, laboratory investigations have revealed that *S. salivarius* engages directly with the gingival epithelium [224], exhibiting anti-inflammatory characteristics and fostering a harmonious immune response [225,226].

Specifically, two strains of *L. reuteri* (DSM 17938 and ATCC PTA 5289) have been shown to significantly reduce halitosis [227]. *L. salivarius* (WB21 and TI 2711) may reduce dental caries through mechanisms including coaggregation and growth inhibition of *S. mutans*, as well as reducing plaque acidogenicity [228]. The probiotic strain *Bifidobacterium animalis subsp. lactis* (*B. lactis*) HN019 showed significant improvement in clinical periodontal indicators, a more efficient reduction in periodontal pathogens, and a decrease in pro-inflammatory cytokine levels within the gingival crevicular fluid [229]. Moreover, *L. plantarum* exhibited notable effectiveness in inhibiting the growth of *S. mutans* and *C. albicans*, disrupting the formation of virulent biofilms, and influencing gene expression related to crucial metabolic pathways, fungal resistance, and oxidative stress. These findings underscore the potential of *L. plantarum* as a promising candidate for controlling cariogenic biofilms and, consequently, for the prevention of dental caries [230].

### 8.2. Peptides in Oral Health and Their Role in Oral Care

Peptides, short chains of amino acids, play critical roles in various biological processes, including promoting health and healing [231]. In the past twenty years, peptides have gained significant attention in the dental field [232,233]. These peptides have been applied across various dental specialties, including endodontic therapy, restorative dentistry, caries control, bone and dental tissue remineralization, and an enhancement in dental materials [232,234]. The integration of peptides in oral care is an emerging area of research with the potential to revolutionize oral health practices.

Antimicrobial peptides (AMPs) such as defensins and cathelicidins are known for their ability to inhibit the growth of oral bacteria, fungi, and viruses [235]. These peptides disrupt the integrity of microbial cell membranes, providing a natural defense against oral pathogens [236]. AMPs exhibit a range of mechanisms to target and inhibit microbes. Primarily, they interact electrostatically with negatively charged microbial membranes due to their cationic nature. This leads to the formation of pores, disrupting membrane integrity and increasing permeability. AMPs allow the leakage of ions, metabolites, and other vital cellular components, ultimately causing microbial cell death. The interfacial activity of AMPs enables them to partition into the membrane–water interfaces, modifying lipid structures. While eukaryotic cell membranes remain unaffected due to their neutral charge, the specific interactions with microbial membranes make AMPs selective and effective in combating pathogens [235].

Naturally occurring peptides in saliva, such as histatins, play a key role in oral health. They possess antimicrobial properties and contribute to maintaining oral tissue integrity, helping to prevent infections and inflammation [237]. Histatins interact with the plasma membrane of pathogens, forming pores that disrupt membrane integrity. This action allows for the leakage of ions such as potassium, as well as other intracellular components, ultimately impairing the pathogen’s cellular functions and leading to cell death. Histatins bind to the Trk1 potassium transporter on yeast cells, causing potassium efflux and further destabilizing the cell’s internal environment. Histatins exhibit both fungistatic (inhibiting fungal growth) and fungicidal (killing fungi) effects, particularly against *Candida albicans* [238]. Among the histatins, Histatin-5 is the most effective, followed by Histatin-3 and then Histatin-1 [239].

Enamel strengthening peptides, such as peptide P11-4, are an innovative development in dental health, designed to support the remineralization and strengthening of tooth enamel [240]. The biomimetic nature of P11-4 allows it to closely resemble the mineral structure of enamel, aiding the regeneration and remineralization of dental hard tissues [240]. Once applied to the tooth surface, P11-4 forms scaffold-like structures with negative-charge domains that attract calcium ions, inducing the precipitation of hydroxyapatite, the mineral essential for tooth enamel [241]. Combined with fluoride, P11-4 enhances the remineralization process by incorporating calcium and phosphate ions into the enamel structure [240]. Studies, including randomized clinical controlled trials, have demonstrated the safety and efficacy of P11-4 in promoting enamel health [240].

### 8.3. Personalized Oral Care Approaches

In addition to probiotics, a comprehensive oral care regimen can support the natural healing of teeth and gums, aligning with the body’s inherent mouth chemistry for enhanced oral health. In this regimen, xylitol is a vital component that helps rebalance mouth acidity. The regular use of xylitol mints and gums has been associated with decreased counts of *S. mutans*, potentially reducing the adherence of plaque [242]. These products consist of xylitol, natural peppermint flavor, magnesium stearate, and natural menthol. Xylitol, being the crucial component, is the one that decreases the presence of *S. mutans*, which is responsible for tooth decay. Xylitol, once taken up by *S. mutans*, significantly disrupts the bacterium’s usual energy production pathways, fundamentally altering its growth and survival mechanisms. Unlike other sugars, xylitol cannot be processed by *S. mutans* for energy, initiating a futile energy cycle that derails the bacterium’s normal metabolic processes. This disruption results in the conversion of xylitol to xylitol-5-phosphate within the bacterial cell, causing the formation of vacuoles and damaging the cell membrane. These alterations impede the bacterium’s growth and viability. Furthermore, the attempt by *S. mutans* to expel the dephosphorylated xylitol-5-phosphate from the cell, with no energy yield from xylitol metabolism, essentially starves the bacterium, hindering its growth and contributing to its eventual death [243].

Following in the regimen is the mouth rinse, comprising stabilized chlorine dioxide (ClO_2_), trisodium phosphate, and citric acid. This distinct formulation highlights its capacity to offer therapeutic advantages, making it an attractive option for patients concerned about taste preferences and tooth discoloration during oral health treatment [244]. ClO_2_, being the compound of interest, acts on volatile sulfur compounds (VSCs), the chief contributors to oral malodor, through a sequential mechanism. Initially, ClO_2_ reacts with thiol compounds like L-cysteine, generating ClO^2−^ and thiol radicals (RS.), subsequently converting thiol compounds into disulfides like cystine. This reduction of thiol compounds progresses through reactions involving the formation of disulfides, addressing malodorous thiols. Simultaneously, ClO_2_ showcases bactericidal properties by permeating bacterial cells and disrupting cytoplasmic amino acids and membrane proteins, leading to microbial cell death. The chemical reactions involved underline the efficacy of ClO_2_ in neutralizing malodor-causing compounds and eradicating odor-producing bacteria, essential for combating oral malodor [245].

## 9. Chemical Reactions

### 9.1. Reaction 1

RSH (e.g., CH_3_SH) + ClO_2_ → RS· + ClO^2·−^ + H^+^

This reaction includes thiol compounds (e.g., L-cysteine), chlorine thiol, chlorine dioxide, hydrogen, and a dioxide radical ion.

### 9.2. Reaction 2

2RS· → RSSR (e.g., CH_3_SSCH_3_)

This reaction includes thiol radical disulfides (e.g., cystine).

### 9.3. Reaction 3

4RSH + ClO^2·−^ → 2RSSR + Cl^−^ + 2H_2_O

This reaction includes thiol compounds, chlorine dioxide, disulfides, chloride, and water.

The reaction also includes compounds (e.g., L-cysteine) and radical (e.g., cystine) ions.

The results from Shinada et al.’s study [245] confirm that a seven-day regimen using a ClO_2_ mouth rinse significantly reduced morning oral malodor and lowered VSCs in healthy subjects. Additionally, it effectively reduced plaque, tongue coating, and *F. nucleatum* counts in the saliva.

Alternatively, individuals can incorporate a mouth rinse into their oral care routine, which includes thymol, eucalyptol, menthol, and methyl salicylate dissolved in ethanol (27%) [246]. Thymol, derived from Thyme essential oil, is an essential component in antiseptic mouthwashes due to its potent antifungal properties. Its mechanism involves altering the membrane lipids’ fluidity and permeability, disrupting the production of hyphae, which are essential structures for the pathogenicity of *C. albicans*. Additionally, thymol significantly reduces the adhesion of *C. albicans* to epithelial cells, an initial step in infection development [247]. Eucalyptol, a primary constituent of Eucalyptus essential oil, features prominently in oral hygiene products due to its potent antimicrobial properties. Its efficacy extends to combating various oral pathogens like *P. gingivalis* and *S. mutans*, commonly associated with periodontitis and dental issues. Eucalyptol operates by disrupting bacterial cell membranes, notably the cytoplasmic membrane. Components like 1.8-cineole and terpinolene modify membrane permeability, compromising its integrity and allowing the increased passage of ions and molecules, leading to cellular dysfunction and eventual bacterial demise [248]. The combined action of these ingredients aims to support oral health by targeting bacteria and promoting a clean and refreshed feeling in the mouth after use [249].

A fluoride toothpaste is an essential part of oral care, featuring sodium fluoride among other vital components like hydrated silica, cellulose gum, and glycerin. These ingredients collectively contribute to fortifying tooth enamel, preventing cavities, and promoting oral hygiene. Fluoride’s action in preventing tooth decay involves multiple mechanisms and chemical reactions. Fluoride ions (F^−^) can replace hydroxyl ions (OH^−^) in the hydroxyapatite crystal structure of enamel, forming fluorapatite (FAP) [250]. This substitution creates a more acid-resistant structure, making the enamel less susceptible to acids produced by plaque bacteria.

The chemical reaction can be represented as follows:Hydroxyapatite (HA) + Fluoride ions (F^−^) → Fluorapatite (FAP) + Hydroxyl ions (OH^−^)

Fluoride deposition is higher in the outer enamel layers, enhancing their acid resistance and conferring caries resistance to the tooth. In an acidic environment, fluoride can form hydrogen fluoride (HF), an undissociated weak acid that can penetrate bacterial cell membranes. HF’s penetration into bacterial cells leads to two physiological impacts: interference with fluoride-sensitive enzymes and intracellular acidification, hindering essential enzymes necessary for cell growth. Fluoride, when present, combines with calcium to form calcium fluoride (CaF_2_) on plaque and enamel surfaces. CaF_2_ acts as a reservoir for fluoride. When the pH drops due to acid, fluoride and calcium ions are released into plaque fluid. Fluoride then diffuses into enamel pores, forming FAP. FAP is more acid-resistant than hydroxyapatite, reducing the demineralization and enhancing the remineralization of enamel between pH 4.5 and 5.5. This process shortens the demineralization period and helps in maintaining enamel health [250,251]. Combinations such as using toothpaste containing fluoride concentrations ranging from 1100 ppm to 1500 ppm, either on its own or paired with a 0.05% sodium fluoride mouth rinse, have demonstrated considerable effectiveness in reducing the progression of root caries [252].

Fluoride has long been relied upon for its role in preventing dental cavities, yet the growing concerns regarding public acceptance and the potential risk of fluorosis in children highlight the need for alternative solutions. Toothpastes incorporating hydroxyapatite (HA) have emerged as fluoride-free options and have demonstrated promising efficacy as anti-cavity agents in recent research [253]. HA toothpaste works by replenishing lost minerals in enamel and restoring its structure. It binds to damaged enamel, filling in surface irregularities and promoting deeper penetration into lesions for comprehensive repair. Mimicking natural enamel, HA particles effectively remineralize teeth, offering an alternative to fluoride-based products [253].

Cetylpyridinium chloride (CPC), a quaternary ammonium compound, is a prevalent antiseptic found in various oral care formulations like mouthwashes and toothpaste [254]. CPC serves as a potent antimicrobial in oral care by engaging in a multifaceted role in bacterial membranes and cellular functions, ensuring its efficacy in combating plaque-forming bacteria for maintaining oral hygiene. Its positive charge enables interaction with negatively charged bacterial membranes, disrupting the balance of ions like Mg^2+^ and Ca^2+^ and integrating its hydrophobic tail into the lipid bilayer. At lower concentrations, CPC disrupts bacterial osmoregulation, causing the leakage of vital cellular components and potentially activating autolytic enzymes. Higher concentrations lead to extensive membrane disintegration, resulting in the leakage of cytoplasmic contents and damage to proteins, nucleic acids, and cell walls. CPC’s effectiveness extends to Gram-negative bacteria, bypassing their complex cell walls due to its molecular size and leveraging damaged cell walls to enhance its penetration. Additionally, CPC’s surfactant properties facilitate the uniform coverage of irregular surfaces, aiding in biofilm disruption [254].

Clove oil, primarily composed of eugenol, eugenyl acetate, carvacrol, and other compounds, has gained attention in dental health due to its therapeutic properties. Clove oil demonstrates potent antibacterial effects against multi-resistant *Staphylococcus* species. Its action inhibits the growth and proliferation of these bacteria, aiding in controlling oral infections. With its antifungal activity, clove oil impedes the growth of fungi like *C. albicans*. It reduces ergosterol levels, a crucial component of fungal cell membranes, and inhibits germ tube formation by *C. albicans*, essential for its pathogenicity. The oil exhibits strong radical scavenging activity, indicating its antioxidant potential. This property helps combat oxidative stress in the oral cavity, contributing to overall oral health [255]. In dentistry, clove oil finds application in various forms, including mouth rinses or gels. Its mode of action involves disrupting microbial cell membranes, inhibiting essential cellular processes, and interfering with the structural integrity of pathogens. These actions collectively contribute to its efficacy in preventing and managing oral infections and inflammation.

Oil pulling is a traditional oral hygiene technique involving swishing oil, typically coconut, sesame, or sunflower oil, in the mouth for around 15–20 min [256]. Oil pulling generates antioxidants that damage the cell walls of microorganisms, killing them effectively. The oils attract the lipid layer of bacterial cell membranes, causing them to stick to the oil and eventually get removed from the oral cavity. The oil gets emulsified during the swishing process, increasing its surface area and efficacy. The oil coats the teeth and gums, hindering bacterial coaggregation and plaque formation. It effectively removes plaque-building bacteria responsible for various oral issues like dental caries, gingivitis, and bad breath. Coconut oil, particularly due to its lauric acid content, exhibits potent antimicrobial properties. It effectively fights *Streptococcus mutans* and *Candida albicans* in biofilm models. The lauric acid in coconut oil reacts with the saliva’s alkalis to form a substance that reduces plaque adhesion and accumulation, aiding in cleansing the oral cavity. Lauric acid also prevents dental caries, possesses anti-inflammatory properties, and promotes overall oral health. Its pleasant taste adds to its appeal [256,257] (Table 3).

o-Cymen-5-ol exhibits antibacterial qualities without adversely altering the oral microbiome, making it an important component of oral care [258]. It has been discovered to specifically target possible oral pathogens while encouraging the development of advantageous bacteria like Rothia, which helps control blood pressure [258]. Furthermore, it has been demonstrated that o-Cymen-5-ol increases the efficacy of colistin against multidrug-resistant Klebsiella pneumoniae without producing cell toxicity or resistance [259]. According to recent research, o-cymen-5-ol and zinc salts together can effectively lower the development of plaque, gingival inflammation, bleeding gums, and halitosis in toothpastes and mouthwashes [260,261]. It is already known that the o-cymen-5-ol/zinc system directly inhibits oral infections such *Fusobacterium nucleatum*, *Actinomyces viscosus, Porphyromonas gingivalis*, *Streptococcus mutans*, and *Candida albicans* [261]. Moreover, o-Cymen-5-ol prolongs substantivity when paired with cetylpyridinium, which makes it a viable substitute for conventional antimicrobials in mouthwash formulations like triclosan and chlorhexidine [262].

Practicing daily dental care (Appendix A ([263,264,265,266,267,268,269,270,271])) with the above-mentioned key components and strategies may ensure the removal of plaque, resolution of gingivitis, prevention of cavities, treatment of periodontal disease, and management of gum issues, all while fostering a harmonious and healthy oral microbiome.

## 10. Conclusions

In conclusion, the oral microbiome is a diverse and complex ecosystem that plays a significant role in maintaining oral and systemic health. Comprising a plethora of bacterial, fungal, and archaeal species, the oral microbiome’s composition and dynamics are intricately linked to various physiological processes. Disruptions in this delicate balance, leading to oral dysbiosis, can precipitate a spectrum of oral ailments like dental caries, gingivitis, periodontitis, halitosis, and oral candidiasis, while also exerting profound effects on systemic health, including gastrointestinal disorders, cardiovascular diseases, endocrine imbalances, and neurological conditions. The interplay between the oral and gut microbiomes, underscored by microbial translocation, emphasizes the bidirectional relationship between oral and systemic health, wherein disturbances in either ecosystem can cascade into the other, exacerbating disease states. Recent research illuminating connections between specific oral bacteria, such as *P. gingivalis*, and systemic diseases like IBD, cancer, Alzheimer’s, and RA, bring out the far-reaching impact of oral health on overall health outcomes.

As our understanding of the oral microbiome expands, it becomes increasingly evident that fostering a balanced oral microbial ecosystem is paramount for holistic health. Embracing evidence-based dental care practices, such as integrating dental probiotics like *S. salivarius* and *Lactobacillus* strains, alongside utilizing oral ingredients like xylitol, fluoride toothpaste, and essential-oil-based mouth rinses, presents a comprehensive approach to oral health maintenance. One could test for these problems and arrive at a personalized dental care plan to address specific needs. These interventions, rooted in scientific validation, offer promising avenues for preventing oral dysbiosis and tackling common oral health issues like bad breath, gum inflammation, and dental caries. By adopting these strategies, individuals can actively contribute to nurturing a balanced oral microbiome, thus safeguarding not only their oral health but also their systemic well-being. Researchers should conduct long-term studies to track changes in the oral microbiome over time and map out the entire oral microbiome in future investigations, including less-studied archaea and fungi. Finding biomarkers for individualized dental treatment and comprehending how mouth microorganisms impact general health are crucial. To develop new health strategies, we must evaluate novel oral health products and apply methods from other disciplines to increase our understanding of the oral microbiota and its impacts.

## Figures and Tables

**Figure 1 microorganisms-12-01797-f001:**
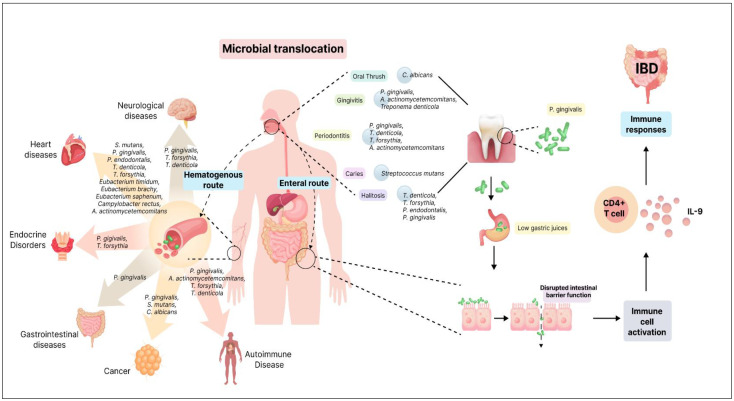
Oral–Gut pathways to IBD and systemic diseases: Oral microbes such as *Porphyromonas gingivalis* can take two distinct routes to enter the body. The first is the hematogenous route (shown on the left), often associated with dental problems like tooth decay, periodontitis, gingivitis, oral thrush, and halitosis, which can further lead to systemic conditions such as CVD, neurological disorders, autoimmune diseases, diabetes, IBD, and cancer. On the other hand, the enteral route (shown in the right) allows these oral microbes to travel from the stomach to the intestines. *P. gingivalis*, which possesses resistance to antibiotics and can survive stomach acid, moves from the stomach into the gut. Changes in gastric acidity can alter the gut microbiota, making it resemble the oral microbiome. Upon entering the gastrointestinal tract, *P. gingivalis* disrupts the intestinal barrier, compromising gut integrity. This disruption, along with changes in the microbiome, initiates inflammation, typically occurring in the ileum. Within this inflammatory response, specific immune cells known as IL9+ CD4+ lamina propria T cells become active and produce IL-9, a cytokine that fuels immune responses and inflammation. While inflammation serves as a defense against invaders, excessive or chronic inflammation can lead to conditions like IBD. Abbreviations: IL-9: Interleukin-9; IL9+ CD4+ lamina propria T cells: immune cells producing IL-9; IBD: inflammatory bowel disease.

**Figure 2 microorganisms-12-01797-f002:**
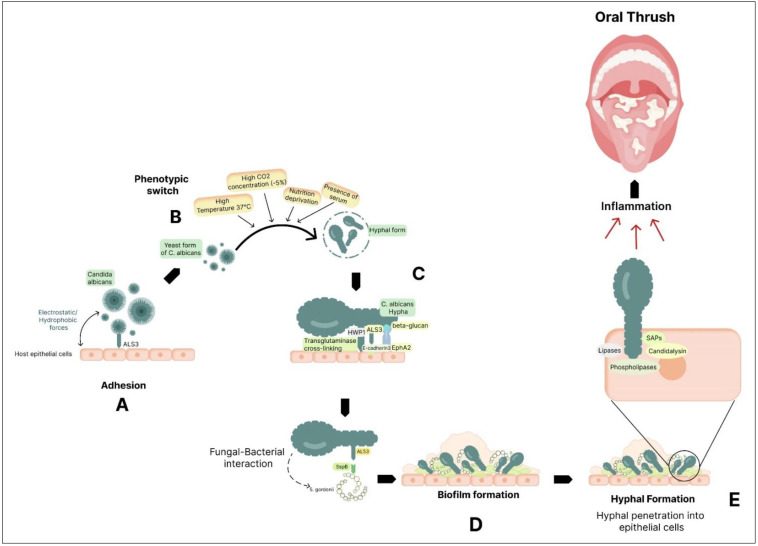
Mechanisms of *Candida albicans* pathogenicity. (**A**) Adhesion *to host surfaces: Candida albicans* initially adheres to host surfaces through weak and reversible interactions influenced by hydrophobic and electrostatic forces, facilitated by host tissue receptor glycoproteins ALS and HWP1. (**B**) *Phenotypic switch: C. albicans* can phenotypically switch between yeast and hyphal states. (**C**) *Host cell receptor interaction: C. albicans* can adhere to epithelial cells using various host cell receptors, such as EphA2 (through β-glucan) and E-cadherin (through ALS3). Host cell transglutaminases create a cross-linking between *C. albicans* and the epithelial surface by interacting with HWP1. (**D**) *Biofilm formation:* Interaction with oral bacteria, such as *Streptococcus gordonii*, promotes the formation of biofilms. ALS3 on *C. albicans* binds to surface protein SspB on *S. gordonii*, facilitating biofilm formation. (**E**) *Hyphal penetration into epithelial cells:* The shift to the hyphal form is crucial for tissue invasion. Hyphae infiltrate and harm epithelial cells, causing inflammation and white patches in oral thrush. Abbreviations: ALS: agglutinin-like sequence; HWP1: hyphal wall protein; SspB: surface protein SspB; EphA2: Eph receptor A2.

**Figure 3 microorganisms-12-01797-f003:**
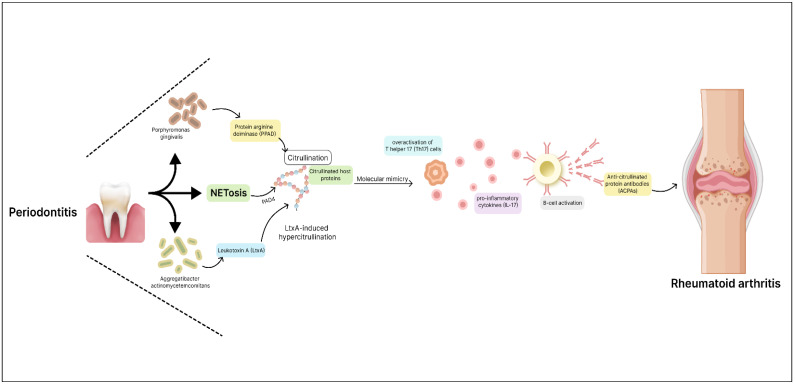
Mechanisms linking oral pathogens to RA. *Porphyromonas gingivalis* and *RA* link: *P. gingivalis* possesses virulent factors, including the PPAD, which citrullinates host proteins. This citrullination is thought to trigger autoimmune responses, leading to the production of ACPAs characteristic of RA. *NETs* and *ACPA* production*:* NETs formed during periodontitis contribute to protein citrullination catalyzed by PAD4 enzymes, further enhancing ACPA production. *Aggregatibacter actinomycetemcomitans* and *RA: A. actinomycetemcomitans* secretes LtxA, triggering neutrophil hypercitrullination and the release of citrullinated autoantigens, further fueling the ACPA response. Citrullinated autoantigens can trigger an autoimmune response, wherein Th17 cells in the immune system become overactivated. Th17 cells are known for producing pro-inflammatory cytokines, notably IL-17. IL-17 stimulates B cells to produce antibodies, including ACPAs. These antibodies can target citrullinated proteins, contributing to the autoimmune processes and inflammation seen in RA. Abbreviations: PPAD: protein arginine deiminase of *P. gingivalis*; ACPAs: anticitrullinated protein antibodies; PAD4: peptidylarginine deiminase 4; LtxA: leukotoxin A; Th17: T helper 17; IL 17: interleukin-17; RA: rheumatoid arthritis.

**Table 1 microorganisms-12-01797-t001:** The composition of the oral microbiome.

Microbial Component	Predominant Phyla/Genera	Other Notable Genera
Bacteria	*Actinobacteria*, *Bacteroidetes*, *Chlamydia*, *Euryarchaeota*, *Fusobacteria*, *Firmicutes*, *Proteobacteria*, *Spirochaetes*, and *Tenericutes*	- Gram-positive cocci: *Abiotrophia*, *Peptostreptococcus*, *Streptococcus*, and *Stomatococcus* - Gram-positive rods: *Actinomyces*, *Bifidobacterium*, *Corynebacterium*, *Eubacterium*, *Lactobacillus*, *Propionibacterium*, *Pseudoramibacter*, and *Rothia* - Gram-negative cocci: *Moraxella*, *Neisseria*, and *Veillonella* - Gram-negative rods: *Campylobacter*, *Capnocytophaga*, *Desulfobacter*, *Desulfovibrio*, *Eikenella*, *Fusobacterium*, *Hemophilus*, *Leptotrichia*, *Prevotella*, *Selemonas*, *Simonsiella*, *Treponema*, and *Wolinella*
Fungi	*Candida*	*Cladosporium*, *Aureobasidium*, *Saccharomycetales*, *Aspergillus*, *Fusarium*, and *Cryptococcus*
Archaea	*Euryarchaeota*	-
Eukaryotic virus	*Herpesviridae*, *Papillomaviridae*, *Anelloviridae*, and *Redondoviridae*	
Bacteriophage	*Siphoviridae*, *Myoviridae*, and *Podoviridae*	

**Table 2 microorganisms-12-01797-t002:** Oral microbes and disease associations.

Disease	Commensals	Pathogens	Directionality	Mechanisms	References
Oral diseases					
Dental caries	*Lactobacillus* spp., *Veillonella*, *Propionibacterium*, *Bifidobacterium*, *Corynebacterium*, and *Capnocytophaga*	*S. mutans*	↑	Acidogenic bacteria produce acidic by-products, leading to demineralization and cavitation; dysbiosis-driven disorder.	[83]
Gingivitis	*Streptococcus* sp., *Actinomyces* sp., and *Veillonella* sp.	*P. gingivalis*, *T. denticola*, *A. actinomycetemcomitans*, *Fusobacterium* sp., and *P. intermedia*	↑	Caused by the accumulation of microbial plaque on the tooth surface, which penetrates the gingival tissue and leads to inflammation.	[97]
Periodontitis	*Prevotella melaninogenica*	*P. gingivalis*, *T. denticola*, *T. forsythia*, *F. nucleatum ss. polymorphum*, and *P. intermedia*	↑	Pathogenic bacteria induce inflammation, oxidative stress, immune activation, and tissue damage; potential systemic implications.	[103]
Halitosis	*Prevotella melaninogenica*, *Veillonella* spp., *Peptostreptococcus*, *Actinomyces* spp., *Eubacterium*, *Megasphaera*, *Selenomonas*, *Leptotrichia*, and *Eikenella corrodens*	*Treponema denticola*, *P. gingivalis*, *P. endodontalis*, and *Tannerella forsythia*	↑	Anerobic bacteria produce VSCs causing malodor; bacterial degradation of sulfur-containing amino acids.	[107,108]
Taste impairment	*Lactobacilli*	-	↑	High levels of acid produced by the bacteria impair taste, affecting taste perception.	[116]
Burning mouth syndrome (BMS)	*Streptococcus*, *Rothia*, *Bergeyella*, and *Granulicatella*	-	↑	Alteration in bacterial strains may contribute to the development of BMS influencing pathways involved in inflammation, immune responses, and sensory perception.	[125]
Oral thrush	*Candida parapsilosis*, *Candida krusei*, and *Candida tropicalis*	*Candida albicans*, *Candida glabrata*, *Candida dubliniensis*, and *Candida guilliermondii*	↑	Overgrowth of *Candida* species due to factors like poor oral hygiene, weakened immune system, or underlying medical conditions; adhesion to host surfaces and tissue invasion through various mechanisms; hyphal formation, biofilm production, and secretion of enzymes that degrade host immune factors.	[127]
Systemic diseases					
IBD	-	*P. gingivalis* and *F. nucleatum*	↑	Oral-resident bacteria may infiltrate the gut microbiome; gut dysbiosis due to altered gut epithelial permeability.	[148]
Atherosclerosis	*Prevotella nigrescens* and *Parvimonas micra*	*S. mutans*, *P. gingivalis*, *P. endodontalis*, *T. denticola*, *T. forsythia*, *Prevotella intermedia*, *Aggregatibacter actinomycetemcomitans*, *Eubacterium timidum*, *Eubacterium brachy*, *Eubacterium saphenum*, and *Campylobacter rectus*	↑	Oral dysbiosis triggers local inflammation, systemic inflammatory responses, oxidative stress, immune activation, and platelet aggregation.	[155,156]
Diabetes	*Capnocytophaga*	*P. gingivalis* and *T. forsythia*	↑	Hyperglycemia and oxidative stress create a conducive environment for microbial dysbiosis; elevated inflammatory reactions.	[172]
Obesity	*Proteobacteria*, *Chloroflexi*, and *Firmicutes*	-	↑	Potential migration of oral bacteria to the gut; alterations in bacterial populations impact metabolic homeostasis.	[179,180]
AD	-	*P. gingivalis*, *T. forsythia*, and *T. denticola*	↑	Oral microbiota’s production of inflammatory agents potentially contributes to neuroinflammation and AD progression.	[186,187]
PD	-	*P. gingivalis*	↑	*P. gingivalis* infection is correlated with PD, with studies demonstrating the presence of gingipain R1 (RgpA) in the bloodstream, indicating systemic dissemination. *P. gingivalis* may contribute to PD pathogenesis by inducing systemic inflammation, promoting hypercoagulability, and exacerbating neurodegeneration.	[197]
RA	*Prevotella*, *Veillonella*, and *Lactobacillus salivarius*	*P. gingivalis* and *A. actinomycetemcomitans*	↑	*P. gingivalis* implicated in RA onset via citrullination and ACPA production; dysbiosis of oral microbiota exacerbates joint inflammation.	[205]
SLE	*Veillonella*, *Streptococcus*, and *Prevotella*	*T. forsythia* and *T. denticola*	↑	Oral microbial dysbiosis and periodontitis may exacerbate SLE via immune activation; potential contribution to autoimmune responses.	[214,215].
Cancer	*S. oralis*, *S. mitis*, *S. sanguinis*, *Lactobacillus fermentum*, *Lactobacillus acidophilus*, and *Bifidobacterium adolescentis*	*P. gingivalis* and *F. nucleatum*	↑	Specific bacteria infiltrate cells, initiate tumor development, and produce cancer-promoting substances like lipopolysaccharides.	[221]

**Table 3 microorganisms-12-01797-t003:** Summary of oral health compounds: mechanisms of action and formulations.

Component	Role/Mechanism	Formulation	References
Probiotics	Support balance of oral microbiota, produce specialized metabolites for maintaining microbiota equilibrium, and promote healthy immunity	Probiotic supplements containing strains such as *Streptococcus salivarius* M18, *Streptococcus salivarius* K12, *Lactobacillus plantarum*, *Bifidobacterium lactis*, *Lactobacillus reuteri*, and *Lactobacillus salivarius* help maintain oral microbiota balance, combat bad breath and gum inflammation, and enhance immune responses	[227]
Xylitol	Rebalances mouth acidity, reduces *S. mutans* counts, and disrupts *S. mutans* energy production	Xylitol mints and gums contain xylitol, natural peppermint flavor, magnesium stearate, and natural menthol	[243]
Chlorine dioxide mouth rinse	Neutralizes volatile sulfur compounds (VSCs), kills odor-producing bacteria, and reduces plaque and *F. nucleatum* counts	Stabilized chlorine dioxide (ClO_2_), trisodium phosphate, and citric acid; typically found in commercially available mouth rinses	[245]
Thymol mouth rinse	Antifungal properties; disrupts *C. albicans* hyphae production and adhesion to epithelial cells	Thymol, eucalyptol, menthol, and methyl salicylate dissolved in ethanol (27%); available in various commercial mouthwash formulations	[247]
Fluoride toothpaste	Strengthens enamel, prevents cavities, and promotes oral hygiene	Sodium fluoride, hydrated silica, cellulose gum, and glycerin, available in various brands and formulations of toothpaste	[222]
Hydroxyapatite toothpaste	Replenishes lost minerals in enamel and restores enamel structure; it is an alternative to fluoride-based products	Hydroxyapatite particles, available in various brands and formulations of toothpaste	[253]
Cetylpyridinium chloride (CPC)	Has antimicrobial properties, disrupts bacterial membranes and cellular functions, and is effective against plaque-forming bacteria	Found in various oral care formulations such as mouthwashes and toothpaste	[254]
Clove oil	Has antibacterial and antifungal effects, inhibits multi-resistant *Staphylococcus* species growth, and reduces *C. albicans* ergosterol levels	Clove oil containing eugenol, eugenyl acetate, and carvacrol, available in various oral care products such as mouth rinses and gels	[255]
Oil pulling	Generates antioxidants to damage microbial cell walls, removes bacteria by attracting them to oil, and hinders bacterial coaggregation and plaque formation	Coconut, sesame, or sunflower oil, typically used as a standalone oral hygiene technique	[256,257]
Antimicrobial peptides	Antimicrobial peptides (AMPs) such as defensins and cathelicidins disrupt microbial cell membranes through electrostatic interactions, forming pores that lead to ion leakage and cell death, providing natural defense against oral pathogens	Antimicrobial peptides (AMPs) like defensins and cathelicidins are integrated into oral care products to combat oral bacteria, fungi, and viruses	[239,240]
Enamel strengthening peptides	Enamel strengthening peptides, exemplified by peptide P11-4, mimic the mineral structure of enamel; they facilitate remineralization by attracting calcium ions, promoting the formation of hydroxyapatite essential for enamel regeneration and strengthening	P11-4: dental gels, pastes, and some toothpastes	[240]

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
