# Peer review of "Oral Microbiome: A Review of Its Impact on Oral and Systemic Health"

_microorganisms, 2024, doi:10.3390/microorganisms12091797_

Round 1

Reviewer 1 Report

Comments and Suggestions for Authors

The authors present an extensive compilation of problems encountered in dentistry, corroborated with diseases, risk factors. Some alternative therapies are also described.

The article has some major problems:

For a review, it was necessary to describe what literature the authors researched, what keywords they used, what the inclusion and exclusion criteria were.

In the history part, the authors jump from ancient Egypt to the last decades. If you keep referring to history, why is nothing being said about the period in which these oral diseases appear to develop more?

The chapter on oral dysbiosis is treated superficially

In treatments chapter, I observed many exclusions

About personalized oral care approaches why o-Cymen-5-ol or bacterial lysates are not mentioned – for example

In the bibliography - (I don't know if it complies with MDPI requirements), why was it preferred not to write the article authors and to cite only from PUBMED site - only the abstracts were read, not the full articles? And of the articles cited in this format, over 70% are over 10 years old.

minor issue

the words in full must be put for the first time and in brackets the abbreviated ones - see NGS on lines 75-79 etc

all bacterial strains must be written in italics - see line 116 etc

Author Response

  1. Summary

  1. Point-by-point response to Comments and Suggestions

Comment 1: For a review, it was necessary to describe what literature the authors researched, what

keywords they used, what the inclusion and exclusion criteria were.

Response 1: We appreciate the feedback. We have now included a detailed description of the literature we researched, the keywords used, and the inclusion and exclusion criteria in the revised manuscript. [This change can be found on lines 108-128.]

Comment 2: In the history part, the authors jump from ancient Egypt to the last decades. If you keep referring to history, why is nothing being said about the period in which these oral diseases appear to develop more?

Response 2: We appreciate the reviewer's observation. We have included additional historical context to cover the period between ancient Egypt and the recent decades to provide a more comprehensive overview of the development of oral diseases over time. [This change can be found on lines 61-68.]

Comment 3: The chapter on oral dysbiosis is treated superficially

Response 3: Thank you for your feedback regarding the chapter on oral dysbiosis. I/We have thoroughly reworded the content to enhance clarity and depth. [This change can be found on lines 331-350.]

Comment 4: In treatments chapter, I observed many exclusions

Response 4: Thank you for your observation regarding the exclusions in the treatments chapter. The treatments highlighted represent a limited set, focusing on those that are most commonly available and relevant. We believe that we have provided a comprehensive list of the most pertinent treatments in this section. However, if there are specific treatments you feel should be included, we would greatly appreciate your input.

Comment 5: About personalized oral care approaches. Why o-Cymen-5-ol or bacterial lysates are not mentioned – for example

Response 5: We thank the reviewer for the suggestion. In our discussion on personalized oral care approaches, we have touched upon o-Cymen-5-ol's antimicrobial properties, which help maintain oral hygiene and reduce plaque formation. [This change can be found on lines 1163-1173]. However, we did not mention bacterial lysates due to limited studies on their use in personalized oral care, though they show promise in modulating the oral microbiome and enhancing immune responses.

Comment 6: In the bibliography - (I don't know if it complies with MDPI requirements), why was it preferred not to write the article authors and to cite only from PUBMED site - only the abstracts were read, not the full articles? And of the articles cited in this format, over 70% are over 10 years old.

Response 6: Thank you for your feedback. We have updated the referencing style to APA as per your suggestion. [This change can be found on lines 1264-1926].

We acknowledge the predominance of older references in our bibliography. This is partly due to the foundational nature of many earlier studies in our field.

Comment 7: The words in full must be put for the first time and in brackets the abbreviated ones - see NGS on lines 75-79 etc.

Response 7: I/we have ensured that all abbreviations are now introduced with their full terms followed by the abbreviation in parentheses on first mention. [This change can be found on lines 83-85]

Comment 8: All bacterial strains must be written in italics - see line 116 etc.

Response 8: Thank you for pointing this out. I/we agree with this comment and the organism names have been changed to italics. [This change can be found on lines 143,144 and on lines 149,150].

Reviewer 2 Report

Comments and Suggestions for Authors

The article is interesting and well-constructed; however, it is very lengthy. The authors should avoid repetitive sections to make the reading more fluid and focus on the most relevant aspects within the context. Additionally, the authors use several tables with a lot of repeated information already included in the text. It would be beneficial to simplify the information in the tables or omit them when necessary. Since this is a narrative review, it is important for the authors to include future perspectives on the topic and identify the main gaps, suggesting how they can be addressed in future studies. The illustrations are very good; congratulations to the authors.

Author Response

Comment 1: The article is interesting and well-constructed; however, it is very lengthy. The authors should avoid repetitive sections to make the reading more fluid and focus on the most relevant aspects within the context. Additionally, the authors use several tables with a lot of repeated information already included in the text. It would be beneficial to simplify the information in the tables or omit them when necessary. Since this is a narrative review, it is important for the authors to include future perspectives on the topic and identify the main gaps, suggesting how they can be addressed in future studies. The illustrations are very good; congratulations to the authors.

Response 1:  Thank you for your feedback and for finding the article interesting and well-constructed. We appreciate your suggestions on streamlining the content. However, we believe that the comprehensive nature of the article is essential to provide a thorough understanding of the topic. The detailed sections and tables are included to ensure clarity and support the information presented in the text. We feel that omitting or simplifying these elements might compromise the depth and completeness of the content.

Moreover, we have addressed your suggestion regarding the future perspectives and main gaps, along with suggestions for addressing them in future studies. [This change can be seen on lines 1194-1202]

Reviewer 3 Report

Comments and Suggestions for Authors

The manuscript by John J. Rajasekaran and colleagues provides a comprehensive review of the composition and role of the oral microbiome and its impact on human health. The authors reviewed the published literature and analyzed nearly two hundred articles. The manuscript discusses in detail the prokaryotic and fungal portions of the oral microbiome and briefly describes the interactions between the different parts of the microbiome. Importantly, the manuscript reviews recent publications concerning the relations between the oral microbiome and oral diseases, as well as the features of the microbiota using different target groups. The manuscript design is logical and it contains an interesting discussion of the impressive number of published articles. The figures are of high quality and easy to understand. In addition, the manuscript contains a short and informative highlights section and a pleasant and interesting historical part.

The manuscript addresses an important area of ​​medical microbiology and may be considered for publication in Microorganisms, but despite a detailed analysis of the cellular component of the oral microbiome, it does not provide an overview of the viral component, including eukaryotic viruses and bacteriophages. The authors should fill this gap.

Some minor issues remain disputable, but they can be corrected and improved through discussion.

- I would recommend removing the word "comprehensive" from the title to make it more neutral.

- Lines 106-107 and following - please use italics in bacterial taxa names only where necessary

- Lines 116-117 - please use italics where necessary

- Line 134 - Many other oral inhabitants can produce biofilms. What makes the biofilms produced by Fusobacteria so special?

Author Response

  1. Summary

  1. Point-by-point response to Comments and Suggestions

Comment 1: The manuscript addresses an important area of ​​medical microbiology and may be considered for publication in Microorganisms, but despite a detailed analysis of the cellular component of the oral microbiome, it does not provide an overview of the viral component, including eukaryotic viruses and bacteriophages. The authors should fill this gap.

Response 1: We thank the reviewer for highlighting this gap in our manuscript. We have addressed this issue by briefly discussing the presence of bacteriophages and eukaryotic viruses in the oral microbiome. [This change can be found on Lines 194-210]

Comment 2: I would recommend removing the word "comprehensive" from the title to make it more neutral.

Response 2: Thank you for pointing this out. I/We agree with this comment. Therefore, I/we have decided to remove the word “comprehensive” from the title. [This change can be found on line 2]

Comment 3: Lines 106-107 and following - please use italics in bacterial taxa names only where necessary.

Response 3: We appreciate the feedback and will ensure that bacterial taxa names are italicized correctly and only where necessary. [This change can be found on lines 131-133]

Comment 4: Lines 116-117 - please use italics where necessary

Response 4: Thank you for pointing this out. I/we agree with this comment and the organism names have been changed to italics. [This change can be found on lines 143,144]

Comment 5: Line 134 - Many other oral inhabitants can produce biofilms. What makes the biofilms produced by Fusobacteria so special?

Response 5: I/We recognize the broader spectrum of biofilm producers and their contributions to oral microbial ecology. Therefore I/we have reworded the content for more clarity. [This change can be found on lines 158-160.]

Round 2

Reviewer 1 Report

Comments and Suggestions for Authors

The Authors have addressed all the remarks. The article can be published.

Author Response

We thank the Reviewer for taking the time to review the article and approving the article for publication. 

Reviewer 3 Report

Comments and Suggestions for Authors

The manuscript has become more interesting and has clearly improved. The authors have added important information about the presence of viruses in the oral microbiome. However, issues regarding the formatting of bacterial taxa names have sometimes remained unchanged.

Table 1 - italics should be used only in the names of genera and species

Line 353, 386, 465 - should be “Prevotella spp.”

Line 488 and following - should be “Veillonella spp., Actinomyces spp.”...

Line 564 - Candida spp. - Candida spp.

Please check this issue throughout the manuscript.

Author Response

We thank the reviewer for bringing this inconsistency to our attention. We have reviewed the manuscript and removed italics from ‘spp’ wherever it was mentioned. This includes lines 161, 165, 168, 354, 387, 388, 439, 466, 467, 489, 490, 491, 492, 565, 615, 793, as well as in Table 2 under the Dental caries and halitosis sections.
